# Performance of ChatGPT on Registered Nurse License Exam in Taiwan: A Descriptive Study

**DOI:** 10.3390/healthcare11212855

**Published:** 2023-10-30

**Authors:** Huiman Huang

**Affiliations:** School of Nursing, College of Nursing, Tzu Chi University of Science and Technology, Hualien 970302, Taiwan; hhuiman@ems.tcust.edu.tw; Tel.: +886-3-8572158 (ext. 2780)

**Keywords:** artificial intelligence, registered nurse, nursing graduate

## Abstract

(1) Background: AI (artificial intelligence) chatbots have been widely applied. ChatGPT could enhance individual learning capabilities and clinical reasoning skills and facilitate students’ understanding of complex concepts in healthcare education. There is currently less emphasis on its application in nursing education. The application of ChatGPT in nursing education needs to be verified. (2) Methods: A descriptive study was used to analyze the scores of ChatGPT on the registered nurse license exam (RNLE) in 2022~2023, and to explore the response and explanations of ChatGPT. The process of data measurement encompassed input sourcing, encoding methods, and statistical analysis. (3) Results: ChatGPT promptly responded within seconds. The average score of four exams was around 51.6 to 63.75 by ChatGPT, and it passed the RNLE in 2022 1st and 2023 2nd. However, ChatGPT may generate misleading or inaccurate explanations, or it could lead to hallucination; confusion or misunderstanding about complicated scenarios; and languages bias. (4) Conclusions: ChatGPT may have the potential to assist with nursing education because of its advantages. It is recommended to integrate ChatGPT into different nursing courses, to assess its limitations and effectiveness through a variety of tools and methods.

## 1. Introduction

ChatGPT has been widely applied. ChatGPT is an interactive tool for medical education to facilitate the learning of new concepts and innovations, having various impacts in our daily lives [1,2]. Much like the learning technology employed in neural networks, ChatGPT has undergone extensive training, being utilized in various areas [3,4,5]. The initial response from ChatGPT can serve as a gateway to follow-up questions that organically explore the fundamental knowledge necessary to substantiate the underlying medical rationale [1,2]. ChatGPT is an extensive textual corpus encompassing books, articles, websites, and various other sources of textual information [3,4,5]. The impact of artificial intelligence relies on its ability to enhance individual learning capacities and clinical reasoning skills and facilitate students’ comprehension of intricate concepts in healthcare education [6,7,8]. However, the complete impact it has on education remains uncertain and necessitates additional investigation [5].

ChatGPT (Chat Generative Pre-trained Transformer) can be programmed to generate different perspectives or suggest relevant resources from the prompt [3,4,5]. By utilizing ChatGPT for step-by-step guidance, interactive learning, and feedback, it is possible to enhance students’ skills and knowledge, individual learning capabilities, and clinical reasoning skills and facilitate their understanding of complex concepts in healthcare education [6,7,8]. ChatGPT increases collaborative activities in nursing education [9]. The utility of ChatGPT encompasses its role as an interactive tool for medical education to facilitate the learning process [1,3]. Moreover, it has the potential to revolutionize the virtual learning of students [10]. However, the application of ChatGPT has its constraints, which encompass the possibility of generating biased output, concerns regarding information accuracy, and its potential influence on students’ critical thinking and communication skills, among other factors [7,11,12]. The specific impact of ChatGPT on nursing education remains undetermined at this juncture [13,14]. Despite its potential to enhance educational efficiency, there is presently limited emphasis on its implementation within the realm of nursing education [13,15].

The Registered Nurse License Exam (RNLE) represents a noteworthy source of apprehension among nursing students in Taiwan. It functions as a pivotal evaluative instrument for gauging the knowledge proficiency of individuals aspiring to attain the status of registered nurses. The RNLE is a standardized examination program that inclusively addresses a spectrum of subjects within the corpus of nursing knowledge. It is imperative to meticulously evaluate both the prospective advantages and drawbacks linked with ChatGPT to proactively anticipate and equip for the future landscape of nursing education. The purpose of this study was to understand the performance of ChatGPT on the registered nurse license exam (RNLE) in Taiwan, and to evaluate the advantages and disadvantages of using ChatGPT, in order to a better understanding of the potential impact of ChatGPT in nursing education and offer recommendations for using ChatGPT as a teaching strategy in nursing.

## 2. Background

ChatGPT first emerged in November 2022. It is a system developed by OpenAI, based on a large language model (LLM), and possesses the ability to simulate human language processing [3,4,5]. AI chatbots can generate human-like responses based on the input text they receive [11,16,17]. ChatGPT provides personalized interactions and responses within a few seconds [18]. ChatGPT empowers the customization of educational materials and evaluation instruments, allowing for personalized adaptations that cater to the unique requirements and learning preferences of individual students [19]. The performance of ChatGPT in complex and intellectually demanding tasks in the medical field still requires further empirical research and evidence-based support [12,20].

Although ChatGPT has been used widely, the application of ChatGPT has its limitations, such as the potential for biased output, the accuracy of the information, copyright problems, and the potential impact on students’ critical thinking and communication skills [4,11,12]. ChatGPT has been trained on inadequate datasets, which can lead to the generation of biased or misleading results [12,21]. Ethical issues should be considered [22]. Using AI chatbots can potentially lead to issues of plagiarism, improper referencing, or bias [4,23,24]. Moreover, ChatGPT potentially causes perspectives and information from AI hallucination, which refers to the AI’s tendency to generate the next word based on the training data it had received, including its own previous responses [16,25,26]. Additionally, due to ChatGPT’s knowledge cutoff being in 2021, it may not be equipped to handle or provide information on more recent data or developments [12,16]. All these opinions show that we need to apply evidence-based research to understand the potential shortcomings of implementing ChatGPT in healthcare education [12]. AI chatbots still lack the ability to present statistical analyses, and although they can simulate text for various scenarios, they are unable to generate visual animations [2,10,24].

Without any professional training or reinforcement, ChatGPT demonstrates impressive performance in various medical evaluation tasks [2,27], such as showcasing its natural language processing capabilities in medical question-and-answer tasks [1,28]. It achieved a score of 60.2% in the US Medical Licensing Examination (USMLE), 57.7% in Multiple-choice medical exam questions (MedMCQA), and 78.2% in the medical reading comprehension dataset (PubMedQA) [27,29]. ChatGPT’s medical knowledge and interpretation in the field of medicine are not yet sufficient for practical application [24]. ChatGPT has also constructed the key trends and developments in nursing [30]. However, knowledge development should be based on the theory of Bloom to scaffold the learning process. The theory consists of six major categories: knowledge, comprehension, application, analysis, synthesis, and evaluation [31]. An example of Yale New Haven Hospital (YNHH), which was an early adopter of the Rothman Index—a tool that reflected patient acuity and risk—has now found its performance positively influenced by nursing assessment data, indicating significant potential for nurses to impact patient care [32,33,34].

The RNLE is a vital exam for nursing students, assessing their knowledge and determining their eligibility as registered nurses in Taiwan. It serves as the benchmark for their professional qualification following nursing education. The Taiwan RNLE consists of a total of five subjects, including “basic medical science (BMS)”, “fundamentals of nursing and nursing administration (FNNA)”, “medical and surgical nursing (MSN)”, “maternal and pediatric nursing (MPN)”, and “psychiatric and community health nursing (PCHN)” [35]. Each subject has a maximum score of 100. The overall examination result is determined by computing the average score across all subjects. To pass, a candidate must attain a minimum total average score of 60. Nevertheless, if a candidate receives a score of zero in any subject, they will not qualify for a passing grade [35,36].

ChatGPT possesses the capability to provide immediate feedback and corrections to inquiries, thereby assisting students in comprehending intricate concepts that may be challenging to grasp [37]. However, the utilization of ChatGPT in nursing education brings with it many unknowns. It is imperative to ascertain the effects of ChatGPT on nursing education, as underscored by O’Connor et al. (2022) [13] and Seney et al. (2023) [14]. To proactively anticipate and prepare for the future of nursing education, a comprehensive evaluation of the prospective advantages and risks linked to ChatGPT is essential. Thus, the purpose of this study was (1) to understand the scores achieved by ChatGPT on the Registered Nurse License Exam (RNLE) and compare the pass rate with nursing graduates in Taiwan, and (2) to evaluate the advantages and disadvantages of using ChatGPT.

## 3. Materials and Methods

### 3.1. Study Design

This study utilized a descriptive research approach to evaluate ChatGPT’s performance on the Registered Nurse Licensure Examination (RNLE) in Taiwan, comparing it with the test scores of nursing graduates. The assessment of ChatGPT’s performance in the RNLE was carried out by employing the GPT-3.5 model, which encompasses diverse elements, including input sourcing, encoding methods, and statistical analysis. Additionally, researchers randomly chose ten items for each examination, encompassing a total of 40 items of ChatGPT’s explanations and analyses, to conduct an in-depth evaluation. Random numbers were used to evaluate the explanation of RNLE using the Random-Number Generator of Microsoft V 2.0.4.2.

### 3.2. Participants

The research subject of this study pertains to the examination items of the Registered Nurse License Exam in Taiwan. Typically, this examination is administered semi-annually to nursing graduates in Taiwan. Each examination comprises five distinct subject areas, with each subject encompassing a variable number of multiple-choice questions ranging from 50 to 80, resulting in a total item count of approximately 370 to 400 questions.

The exclusion criteria were RNLE after 2021, due to ChatGPT’s knowledge cutoff being in 2021 [12,16]. This study would have included the exams from 2022 to 2023 of RNLE. However, following feedback from schools, the number of questions was reduced from 80 to 50 in the 2nd exam in 2022 and remained at 50 in 2023 [35,36]. All questions (items) were prompted by research assistants and encoded by the researcher. The quantitative variables were the score and the correct percentage of five subjects on RNLE, as well as the comparable pass percentages of RNLE in Taiwan.

### 3.3. Data Sources/Measurement

The data in question comprise the RNLE items sourced from the Ministry of Examination in Taiwan. The process of data measurement encompasses two distinct phases (Figure 1): (1) input sourcing and (2) encoding.

#### 3.3.1. Input Source

Over the course of four exams spanning from 2022 to 2023, each consisting of five subjects—BMS, FNNA, MSN, MPN, and PCHN—a total of 370 to 400 items were included. The input source encompassed all questions from RNLE in 2022 to 2023, which were then formatted into two sections: one comprising questions and the other featuring multiple-choice single answers.

#### 3.3.2. Encoding

Qualitative data were derived from ChatGPT’s responses to RNLE items, encompassing answers, explanations, and narratives. The total number of items included 400 in the exam of 2022 1st, and 370 in the exam of 2022 2nd and 2023, sequentially arranged from 1 to 400 or 1 to 370. Based on the Random-Number Generator of Microsoft V 2.0.4.2., each exam randomly selected ten items (questions) within five subjects—BMS, FNNA, MSN, MPN, and PCHN. ChatGPT was provided with randomly sampled items as prompts, which were research-encoded, and the responses generated from ChatGPT were analyzed. The responses included answers and explanations.

### 3.4. Statistical Methods

This study used IBM SPSS Statistics 26.0 for data analysis. The quantitative variables were used to analyze the scores and correct rate of RNLE achieved by ChatGPT in Taiwan from 2022 to 2023. The qualitative data were encoded and the questions were examined via random selection, and the advantages and disadvantages of using ChatGPT on RNLE were analyzed. As this study did not involve human or animal subjects, but the analysis of the results of the RNLE in Taiwan, ethical approval and informed consent were not required.

## 4. Results

The aim of this study was to understand the performance of ChatGPT on the registered nurse license exam (RNLE) and evaluate the limitations and advantages of using ChatGPT. According to the Taiwanese Ministry of Examination (2023) [35,36], a passing grade requires a total average score of at least 60. The pass rate for the examination of the RNLE in 2022 1st and 2023 1st was relatively low, standing at 18.17% and 20.42%, compared to the other exams that were 45.31% and 45.73% (Table 1), respectively. The number of candidates in these exams was also totally different, and the second exam obviously had more candidates than the first exam in 2022 and 2023. The number of candidates participating in the RNLE in Taiwan exhibits a significant disparity across the four administrations, particularly with a notably lower turnout observed in the first examination each year compared to the second. Moreover, the pass percentage for the RNLE in Taiwan also demonstrates a conspicuous discrepancy.

The process of data measurement encompasses two distinct phases (Figure 1): (1) input sourcing and (2) encoding. An example of an input source is “The patient is suspected of anaphylactic shock, and the medical order is as follows: adrenalin (epinephrine) 1 mg/mL/amp 0.3 mg IM stat. Which of the following routes of administration is incorrect?”, “Which of the following statements about nursing instructions for medication administration in myasthenia gravis patients is correct?”, or “Miss Zhao has been diagnosed with schizophrenia. Which of the following statements about schizophrenia is correct?” Here is an example of a multiple-choice question with single answers: “(A) Discussing the use of asthma medication with a pharmacist, (B) Performing urinary catheterization as instructed by a physician, (C) Instructing a primipara on breastfeeding, (D) Administering antibiotics every 6 h [38]”.

An example of an encoding and explanation is as follows: “Mr. Wang, who has undergone a colon resection surgery and just returned to the patient ward from the recovery room, has four health problems identified by the nursing staff according to his needs. In terms of priority for handling, which one is ranked last? (A) Ineffective airway clearance; (B) Acute pain; (C) Lack of knowledge about home care; and (D) Changes in peripheral tissue perfusion” [38]. The answer is “(C) While important, it holds a lower priority compared to the preceding issues. Once the patient’s physiological condition stabilizes, the nursing team can provide more education and guidance” [39].

### 4.1. The Performance of ChatGPT on RNLE

ChatGPT underwent testing with four exam questions of 370~400 questions (items) in 2022~2023. The scores were between about 37.50 and 78. Of the four exams, the lowest score was recorded in the exam of 2022 2nd FNNA, with a score of 37.50. Additionally, the exam of MSN in 2023 1st yielded a score of 45.25, and the exam of PCHN in 2023 1st resulted in a score of 42.50. The highest scores were achieved in the exam of BMS in 2023 1st, with a score of 78, as well as in the exam of PCHN in 2022 1st and BMS in 2023 2nd, both of which earned a score of 70. There were significant differences in the scores for all subjects across the four exams from 2022 to 2023 (Table 2). The correct percentage was 54.05~63.75%. The correct percentages were 63.75%, 58.37%, 54.05%, and 60%. ChatGPT performed well in the 1st 2022 and 2nd 2023 exams, with the percentages of 63.75 and 60, respectively. Overall, ChatGPT’s performance in subjects such as FNNA, MSN, and MPN was relatively poor, with particular emphasis on the fact that the average scores for MSN consistently fell below 60 points.

ChatGPT displayed variations in performance across different subject exams in the RNLE, with its most robust performance observed in the 2022 1st compared to the other exams. ChatGPT obtained mean scores ranging from 51.6 ± 10.16 to 63.75 ± 5.22. ChatGPT required a total average score of 63.75 ± 5.22 and 60.75 ± 6.88 in the exam of 2022 1st and 2023 2nd, respectively, and passed the examination. In the four rounds of examinations, two of them resulted in scores exceeding 60 points. In the two examinations for the RNLE, the scores for FNNA and MSN were notably higher compared to the other two rounds of assessments. On the other hand, compared to the scores of the other two exams, the score was below the standard, so it failed. In an attempt to assess ChatGPT’s performance across four consecutive examinations in the field of the RNLE, it becomes apparent that there were significant fluctuations in the average scores for the subjects of BMS, FNNA, MSN, and PCHN from 2022 to 2023. Interestingly, the subject of MPN exhibited smaller variations during the same period, highlighting a degree of stability in its performance (Figure 2).

### 4.2. Evaluation of the Advantages and Disadvantages of Using ChatGPT

A total of forty items were selected via the Random-Number Generator of Microsoft V 2.0.4.2. to examine the answers and explanations of the RNLE by ChatGPT. ChatGPT answered a total of 27 questions correctly with accurate explanations, but it got 13 questions wrong and provided incorrect explanations for them. ChatGPT could answer most questions directly, such as “Question 1” and “Question 2” (Table 3—example 1). After reading the question, ChatGPT did not provide a correct answer but instead explained the feasibility of each answer, the explanations provided for each question were all accurate, and the correct answer rationale could not be discerned, such as “Question 3”. There was no definitive answer to this question (Table 3) [39].

In this current study, the initial response from ChatGPT and the subsequent answer were totally different. ChatGPT would provide incorrect answers and information, for example, “Question 4” (Table 4—example 2), which indicated “During the fourth stage of labor, it is customary for the healthcare team to administer a vitamin K1 injection to the newborn as a preventive measure against hemorrhagic disorders [39]”. “Question 5” (Table 4) indicated that black stools are typically caused by upper gastrointestinal bleeding, rectal bleeding, iron supplementation, and overconsumption of pork liver but generally do not result in black stools. All of the explanations were incorrect based on the concept and nursing knowledge. “Question 6” (Table 4) also demonstrated that ChatGPT relied on a database focused on postoperative care for surgical patients, rather than addressing specific considerations for post-PTCA care. It considered a singular treatment approach or aggregated different medical interventions, which could potentially introduce bias into the answers provided.

Some specific scenarios are as follows: In certain instances, ChatGPT provides different answers to the same question, as exemplified by “Question 7” (Table 5—Example 3). This variability in responses can be attributed to factors such as language differences or varying regulations across different countries when the questions are posed in either Chinese or English. Moreover, in the case of more complex clinical scenario questions or those offering multiple choices, ChatGPT is prone to confusion, making it challenging to determine the correct course of action. This can result in incorrect answers, despite providing a correct explanation of the rationale (Table 5).

## 5. Discussion

In this study, we presented substantiated evidence about the performance of ChatGPT on the RNLE. The findings can be organized into two parts: (1) the score of ChatGPT on the RNLE and (2) the evaluation of the advantages and disadvantages of using ChatGPT.

### 5.1. The Performance of ChatGPT on RNLE

The current investigation utilized a multiple-choice examination format. ChatGPT demonstrated the capability to furnish rapid responses within seconds, as well as to provide answers and information during the interaction, consistent with prior research by Ahn (2023) [9] and Ali et al. (2023) [18]. The findings unveiled that ChatGPT possessed a rudimentary grasp of nursing knowledge, resulting in a success rate exceeding 54.05%. Nevertheless, a comparative analysis of the four examinations revealed that ChatGPT exhibited a lower pass rate and mean scores below 50 for the 2nd quarter of 2022 and the 1st quarter of 2023. The overall average score was 60, and any subject receiving a score of zero would not meet the passing criteria [35,36].

ChatGPT attained an overall average score of 63.75 ± 5.22 in the 2022 1st exam and 60.75 ± 6.88 in the 2023 2nd exam, both of which met the RNLE passing criteria. The higher average scores achieved by Taiwanese candidates may be attributed to their prior acquisition of pertinent knowledge prior to the examination, as noted by Huh (2023) [24]. Huh’s study (2023) [24] revealed that ChatGPT correctly answered 60.8% of questions, which fell short of the 90.8% accuracy achieved by medical students. In the study by Kung et al. (2023) [29], ChatGPT was able to consistently approach or surpass the 60% accuracy threshold, corresponding to the proficiency level of a third-year medical student or a first-year resident [1,28]. Furthermore, in the present study, ChatGPT obtained scores ranging from 48 to 78 for the BMS subject, aligning with the findings of Talan and Kalinkara (2023) [2]. ChatGPT demonstrated exceptional performance, particularly in anatomy examinations.

The pass rates for the RNLE examination in both 2022 and 2023 were relatively low, registering at 18.17% and 20.42%, possibly influenced by factors such as the timing of students’ graduation and their level of preparedness for re-examination, which are rooted in Taiwanese cultural norms. RNLE candidates encompass individuals who graduated in the same year as the examination and those who had previously failed the test. Given that students typically graduate in June each year and subsequently participate in the second round of the RNLE, the number of candidates for the first examination in both 2022 and 2023 was notably lower compared to the second examination.

### 5.2. To Evaluate the Advantages and Disadvantages of Using ChatGPT

ChatGPT 3.5 https://openai.com/blog/chatgpt/ (accessed on 20 September 2023) is convenient and accessible for everyone, and the software is easy to operate. ChatGPT answers most questions directly. The study by Kung et al. (2023) [29] indicated that ChatGPT exhibits a remarkable level of concordance and insight in its explanations. ChatGPT is capable of delivering responses within a short timeframe [2]. However, in the current study, ChatGPT might provide incorrect explanations or information, as observed in the study conducted by Huh (2023) [24]. ChatGPT’s medical knowledge and interpretative capabilities within the field of medicine remained insufficient and unreliable. Ahn (2023) [10] highlighted ChatGPT’s tendency to retrieve erroneous information and skills through web searches, potentially stemming from inaccurate data sources. According to the findings of this study, ChatGPT exhibited the potential to generate misleading or incorrect explanations, even though it adeptly rephrased unclear or incomprehensible questions [2]. As examinations advanced, this could culminate in misinterpretations, particularly when the veracity of certain content generated by ChatGPT became challenging to ascertain [16,25].

The responses generated by ChatGPT in this study exhibited inconsistency between initial replies and subsequent answers. Hallucination may be attributed to ChatGPT’s susceptibility to confusion or misinterpretation of complex scenarios, resulting in divergent explanations and responses. AI hallucination refers to the AI’s tendency to generate the next word based on the training data it had received, including its own previous responses [26]. This phenomenon might be influenced by the constraints placed on the model’s knowledge, particularly those limited to information available before 2021 [12,16]. Additionally, factors contributing to these inconsistencies include the model’s generation diversity, its sensitivity to variations in contextual information, and the inherent uncertainty inherent in the machine learning process [12]. It is worth noting that ChatGPT still faces challenges in formulating appropriate interventions and nursing plans tailored to individual patient needs [2,30]. Furthermore, this study did not assess ChatGPT’s capacity to process image-based questions [18]. Despite AI’s ability to simulate human behavior within specific contexts, it remains incapable of substituting human creativity, critical thinking, and clinical reasoning. Furthermore, Sallam (2023) [12] argued that excluding different languages, such as English records, may introduce selection bias. In the present study, RNLE items were exclusively presented to ChatGPT in Chinese, which may impose certain limitations stemming from a narrower data corpus. The advantages and disadvantages of using ChatGPT in the RNLE are shown in Table 6.

## 6. Limitations

In the current study, researchers only evaluated the performance of ChatGPT using three RNLE assessments conducted after 2021. The sample size RNLE should increase items to find more risks of using ChatGPT. Additionally, the items used to assess ChatGPT were in Chinese, which could potentially introduce selection bias. These factors may impose certain limitations on this study. It could display some different responses between ChatGPT 3.5 and 4.0.

## 7. Conclusions

This study aimed to understand ChatGPT’s performance on the RNLE. ChatGPT displayed a high level of concordance and insight in exams. ChatGPT has the potential to assist with nursing education. Although ChatGPT provides information and knowledge in seconds as this current study has shown, it cannot replace the essential elements of nursing care, such as clinical reasoning and clinical judgment, especially in complex scenarios. The findings suggest that ChatGPT has correctly responded to over 50% of the RNLE. However, based on the explanations of ChatGPT, further investigation is needed to assess its effectiveness in promoting comprehension, application, synthesis, and evaluation, as per Bloom’s taxonomy. Caution should be exercised when using ChatGPT, particularly due to potential inaccuracies in responses and concerns regarding plagiarism in written assignments due to hallucination or misleading answers. We believe ChatGPT would reach a maturity level and impact clinical nursing care. To anticipate and prepare for the future of nursing education, it is essential to thoroughly assess both the potential benefits and risks associated with ChatGPT. In the future, it is recommended to integrate ChatGPT into different nursing courses, to assess its limitations and effectiveness through a variety of tools and methods, and to evaluate the response in different clinical situations and nursing care.

## Figures and Tables

**Figure 1 healthcare-11-02855-f001:**
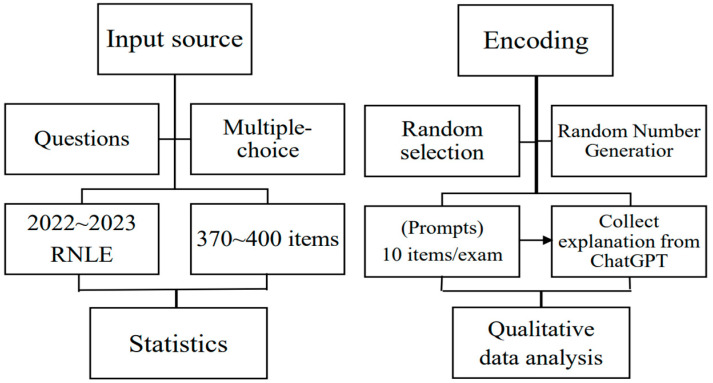
The framework for evaluating ChatGPT’s performance on the RNLE.

**Figure 2 healthcare-11-02855-f002:**
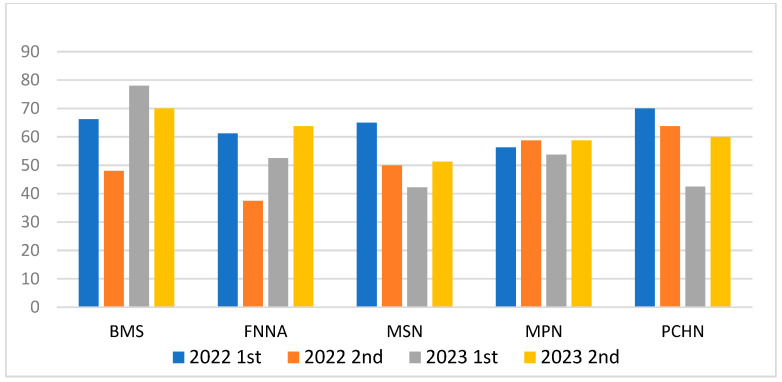
The performance of ChatGPT on registered nurse license exam. Note. BMS = basic medical science; FNNA = fundamentals of nursing and nursing administration; MSN = medical and surgical nursing; MPN = maternal and pediatric nursing; PCHN = psychiatric and community (Ministry of Examination, 2023).

**Table 1 healthcare-11-02855-t001:** The performance of candidates on the registered nurse license exam.

	Exams	2022 1st	2022 2nd	2023 1st	2023 2nd
Number					
Candidates in Taiwan		6873	14,699	7476	13,990
Number of passes		1031	6018	1267	5758
Pass percentage		18.17%	45.31%	20.42%	45.73%

**Table 2 healthcare-11-02855-t002:** The registered nurse license exam scores of ChatGPT.

RNLE Items Subjects *	2022 1st		2022 2nd		2023 1st		2023 2nd	
	Correct/Incorrect	Scores	Correct/Incorrect	Scores	Correct/Incorrect	Scores	Correct/Incorrect	Scores
BMS **	53/27	66.25	24/26	48.00	39/11	78.00	35/15	70.00
FNNA	49/31	61.25	30/50	37.50	42/38	52.50	51/29	63.75
MSN	52/48	65.00	40/40	50.00	37/43	45.25	41/39	51.25
MPN	45/35	56.25	47/33	58.75	43/37	53.75	47/33	58.75
PCHN	56/24	70.00	51/29	63.75	39/41	42.50	48/32	60.00
Correct percentage	63.75%		58.37%		54.05%		60%	
Mean score (SD)	63.75 ± 5.22		51.6 ± 10.16		54.4 ± 4.02		60.75 ± 6.88	

Note. * BMS = basic medical science; FNNA = fundamentals of nursing and nursing administration; MSN = medical and surgical nursing; MPN = maternal and pediatric nursing; PCHN = psychiatric and community health nursing. ** BMS had decreased items from 80 to 50 in 2022 2nd (Ministry of Examination, 2023).

**Table 3 healthcare-11-02855-t003:** The ChatGPT prompts and answers on the registered nurse license exam (Example 1).

Question 1	Which of the following supplies nourishment to the lungs through the bronchial arteries? Answer: (C) (A) Pulmonary artery (B) Pulmonary vein (C) Thoracic aorta (D) Internal thoracic artery [38].
ChatGPT	The answer is (C). The thoracic aorta is responsible for transporting oxygenated blood from the heart to the lungs and supplying the bronchial arteries to nourish lung tissues [39].
Question 2	Mrs. Wang, G1P0, who has had normal prenatal check-ups in the past, presents at 32 weeks of gestation with a complaint of reduced fetal movements and is feeling very concerned. What would be the most appropriate action at this time? Answer: (C) (A) Instruct the patient to return home and record daily fetal movements. (B) Explain that this is a common occurrence at this stage and reassure her not to worry. (C) Schedule a non-stress test. (D) Immediately arrange for an induction of labor [38].
ChatGPT	The answer is (C). This test can help assess the well-being of the fetus by monitoring its heart rate and response to its own movements, providing valuable information about the baby’s condition [39].
Question 3	Regarding the description of the Study of Osteoporotic Fractures frailty index (SOF), which of the following statements is incorrect? Answer: (B) (A) There are three assessment items. (B) Weight loss: A weight loss of more than 5 kg compared to one year ago. (C) Lower extremity function: Inability to rise from a chair without using hands five times. (D) Meeting two or more criteria qualifies as frailty [38].
ChatGPT	No answer. (A), (B), (C), and (D) are all correct. Based on the description, individuals meeting two or more of these three assessment criteria are classified as frail. In summary, according to the provided description, there are no errors. In summary, based on the descriptions provided in the data, there is no definitive answer to this question [39].

**Table 4 healthcare-11-02855-t004:** The ChatGPT prompts and answers on the registered nurse license exam (Example 2).

Question 4	Ms. Chen, who has been in labor for 12 h, has given birth to a healthy newborn. Regarding the nursing interventions performed by the nurse during the fourth stage of labor, which of the following is not appropriate? Answer: (D) (A) Encouraging the mother to have eye-to-eye contact with the baby. (B) Wrapping the baby in a blanket to keep warm and encouraging touching the baby. (C) Delaying the administration of vitamin K1 and initiating skin-to-skin contact first. (D) Placing the newborn in an incubator to allow the mother to rest [38].
ChatGPT	The answer is (C). During the fourth stage of labor, it is customary for the healthcare team to administer a vitamin K1 injection to the newborn as a preventive measure against hemorrhagic disorders. Placing the newborn in an incubator to allow the mother to rest (D) is a reasonable arrangement to ensure that the mother has an opportunity to rest and recover postpartum [39].
Question 5	Which of the following situations is less likely to result in black stools in a patient? Answer: (D) (A) Duodenal ulcer (B) Rectal bleeding (C) Iron supplementation (D) Overconsumption of pork liver [38].
ChatGPT	The answer is (C). Black stools are typically caused by upper gastrointestinal bleeding, and rectal bleeding, iron supplementation, and overconsumption of pork liver generally do not result in black stools. Food undergoes normal digestion, which does not result in black stools [39].
Question 6	Which of the following statements regarding postoperative care for patients undergoing percutaneous transluminal coronary angioplasty (PTCA) for angina is incorrect? (A) Monitor pulses in both lower limbs. (B) Monitor electrocardiogram. (C) Encourage the patient to drink plenty of water. (D) Encourage the patient to perform active lower limb joint exercises. Answer: (D) [38].
ChatGPT	The answer is (C). Encouraging the patient to drink plenty of water is not typically a part of the immediate post-procedure care plan. The other options (A, B, and D) are more relevant to the post-PTCA care: The answer (D), it can help prevent complications such as deep vein thrombosis (DVT) and promote circulation. Option (C) is not a standard postoperative care practice for PTCA patients and is therefore the incorrect statement in this context [39].

**Table 5 healthcare-11-02855-t005:** The ChatGPT prompts and answers on the registered nurse license exam (Example 3).

Question 7	Which of the following statements about home-based care services is less appropriate? Answer: (D). (A) Nursing personnel can independently establish home nursing facilities. (B) Home care can provide care for nasogastric tubes, tracheostomies, and urinary catheters. (C) Home rehabilitation services may be provided when deemed necessary based on an assessment by care management specialists. (D) Home pharmaceutical services are not within the scope of services [38].
ChatGPT-1	The answer is (A). Nursing personnel who wish to independently establish home care facilities must adhere to specific laws and regulations, typically requiring certification and licensing by government authorities. Nursing personnel cannot arbitrarily establish home care facilities but must instead comply with relevant laws and regulations, subject to regulatory oversight [39].
ChatGPT-2	The answer is (D). Home-based care services can include medication management and assistance, so stating that home pharmaceutical services are not within the scope of services is less appropriate [39].
Question 8	According to the Occupational Health and Safety Regulations, how often should an employer arrange a general health examination for employees under the age of 40? (A) Once a year. (B) Once every two years. (C) Once every three years. (D) Once every five years. Answer: (C) [38].
ChatGPT	The answer is (B) in Chinese. However, the answer switch (C) in English. As per the Occupational Health and Safety Regulations, employers are required to arrange a general health examination once every two years. This regulation aims to ensure the well-being and health of the workforce by periodically assessing their general health status and identifying any potential health issues [39].
Question 9	When nurses implement physical restraints on patients, which of the following considerations should be taken into account? ① Patients with excessive salivation should be maintained in a lateral position. ② Patients should be turned after being restrained for more than 2 h. ③ If Haloperidol is administered before restraint, blood pressure changes should be closely monitored. ④ Patients typically resist restraint, and it is not necessary to explain the reasons for the restraint. ⑤ After restraint, patients should be allowed to rest, and further discussions with the patient should be avoided. (A) ①②③. (B) ①③⑤. (C) ②③④. (D) ②④⑤. Answer: (A) [38].
ChatGPT	The answer is (B). Options ④ and ⑤ are not included in the correct answer because: It is generally recommended to explain the reasons for restraints to the patient to ensure their understanding and cooperation. Allowing the patient to rest after restraint is important, but it doesn’t preclude the possibility of further discussions or consultations with the patient [39].

**Table 6 healthcare-11-02855-t006:** Advantages and disadvantages of using ChatGPT.

Advantage	Disadvantage
Convenient and accessible; Software is easy to operate; Respond promptly within seconds; Basic level of nursing knowledge: correctly answered over 50%; ChatGPT exhibited exceptional performance in BMS and PCHN.	Medical knowledge and interpretation are inadequate; Confusion or misunderstanding about complicated scenarios and multiple choices; Hallucinations: information comes from inaccurate and unreliable sources; Languages might induce bias.

## Data Availability

Not applicable.

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
