# Peer review of "Performance of ChatGPT on Registered Nurse License Exam in Taiwan: A Descriptive Study"

_healthcare, 2023, doi:10.3390/healthcare11212855_

Round 1
Reviewer 1 Report
Comments and Suggestions for Authors
This paper shows an interesting analysis of the performance of the use of ChatGpt applied to a nurse licence exam.
The work is mainly descriptive explaining the weaknesses and the strengths of ChatGPT especially in answering to questions related to nursing topics , in order to check if this software is able to pass the exam.
Comments:
1- In the abstract is not well defined what is the impact of this study. Maybe it could be better mention in the abstract the purpose that is explained at row 49.
2 - At row 27 you describe the benefit of AI-chatbot in learning experience. In this section I suggest you to mention the important contribution of AI in education and inserting one or more references for example:
- Nalli G. , Amendola D. , Smith S. Artificial intelligence to improve learning outcomes through online collaborative activities. European Conference on e-Learning . 2022, 21, 475–479.
3- at row 164 you mentioned "Window 26.0", maybe it's a mistake. Please write the correct version of O.S. Windows.
4- at row 166 you wrote that you examined and encoded the limitation and risk. What approach did you use? Can you explain deeply this section?
5- at row 262, you described the limitation and advantages of using ChatGPT. it could be clearer with a Table that highlights the Weaknesses and Strengths. I suggest you to improve this part to be more understandable.
Author Response
Thank you for the opportunity to re-submit our manuscript, titled “Performance of ChatGPT on Registered Nurse License Exam in Taiwan: A Descriptive Study”. We have carefully reviewed the reviewers’ comments and made revisions accordingly. Our response to reviewers’ comments is attached, with the revised portion highlighted in red. Please contact me if you have questions or need additional information. Thank you very much.
|
1. |
In the abstract is not well defined what is the impact of this study. Maybe it could be better mention in the abstract the purpose that is explained at row 49. |
The abstract has been revised. |
|
2 |
At row 27 you describe the benefit of AI-chatbot in learning experience. In this section I suggest you to mention the important contribution of AI in education and inserting one or more references for example: - Nalli G. , Amendola D. , Smith S. Artificial intelligence to improve learning outcomes through online collaborative activities. European Conference on e-Learning . 2022, 21, 475–479. |
The introduction has been revised. |
|
3 |
at row 164 you mentioned "Window 26.0", maybe it's a mistake. Please write the correct version of O.S. Windows. |
The information has been revised in the manuscript. |
|
4 |
at row 166 you wrote that you examined and encoded the limitation and risk. What approach did you use? Can you explain deeply this section? |
The information has been revised in the manuscript. |
|
5
|
at row 262, you described the limitation and advantages of using ChatGPT. it could be clearer with a Table that highlights the Weaknesses and Strengths. I suggest you to improve this part to be more understandable. |
The information has been revised in the manuscript. |
Reviewer 2 Report
Comments and Suggestions for Authors
Introduction:
· Begin with a more explicit hook, possibly illustrating the intersection of AI technology, like ChatGPT, and nursing education.
· Break down long, complex sentences into shorter, more digestible ones to facilitate better understanding.
· Detail the potential impacts and applications of ChatGPT in the medical education field, providing concrete examples where possible.
· Explicitly state the primary focus or argument of the paper in the thesis statement, ensuring that it encompasses the main points to be discussed in the article.
· Utilize consistent terminology and fix minor grammatical issues to maintain a formal and coherent tone throughout.
Background:
· Organize information logically and cohesively, ensuring similar ideas are grouped together to maintain a coherent flow throughout the section.
· Clearly and succinctly define technical terms and provide examples or context where needed, to aid reader understanding.
· Provide more detailed examples of ChatGPT’s practical application in the medical field and its successes, alongside its limitations.
Material Methods and Results:
· This section is perfectly written.
Discussion:
1. Use more formal and precise language such as “presented” instead of “provide,” and “substantiated” instead of “impressed.”
2. Consider rewriting sentences for conciseness and clarity, for instance: “The findings from this study are organized into two main parts: (1) the performance of ChatGPT on RNLE, compared to the pass rate of nursing graduates; and (2) the evaluation of the limitations and advantages of using ChatGPT.”
3. Address inconsistencies in verb tenses, e.g., “ChatGPT may confuse or misunderstand…” instead of “ChatGPT may confusing or misunderstand…”
4. Clearly articulate the implications of ChatGPT’s limitations, offering insights into potential improvements and future research directions.
5. When mentioning other studies, ensure clear and concise integration of their findings, comparing and contrasting effectively with the current study’s results.
Conclusion:
· The first sentence could be made more concise, e.g., “This study aimed to understand ChatGPT's performance on RNLE.”
· Avoid vague phrases like “may have potentially” – instead, be more definite: e.g., “ChatGPT has the potential…”
· Clearly articulate the practical implications and future applications of using ChatGPT in nursing education, ensuring that these are grounded in the study's findings.
· “Using ChatGPT… should be approached with caution, particularly due to hallucination misleading the answer, or regarding issues like plagiarism in article writing and homework assignments,” could be rephrased for clarity and conciseness, e.g., “Caution should be exercised when using ChatGPT, particularly due to potential inaccuracies in responses and concerns regarding plagiarism in written assignments.”
· Proofread for grammar and consistency in verb tense and terminology.
Author Response
Thank you for the opportunity to re-submit our manuscript, titled “Performance of ChatGPT on Registered Nurse License Exam in Taiwan: A Descriptive Study”. We have carefully reviewed the reviewers’ comments and made revisions accordingly. Our response to reviewers’ comments is attached, with the revised portion highlighted in red. Please contact me if you have questions or need additional information. Thank you very much.
|
1 |
Introduction: Begin with a more explicit hook, possibly illustrating the intersection of AI technology, like ChatGPT, and nursing education. Break down long, complex sentences into shorter, more digestible ones to facilitate better understanding. Detail the potential impacts and applications of ChatGPT in the medical education field, providing concrete examples where possible. Explicitly state the primary focus or argument of the paper in the thesis statement, ensuring that it encompasses the main points to be discussed in the article. Utilize consistent terminology and fix minor grammatical issues to maintain a formal and coherent tone throughout. |
The introduction has been revised. |
|
2 |
Background: Organize information logically and cohesively, ensuring similar ideas are grouped together to maintain a coherent flow throughout the section. Clearly and succinctly define technical terms and provide examples or context where needed, to aid reader understanding. Provide more detailed examples of ChatGPT’s practical application in the medical field and its successes, alongside its limitations.
|
The background has been revised. |
|
3 |
Discussion: 1.Use more formal and precise language such as “presented” instead of “provide,” and “substantiated” instead of “impressed.” 2.Consider rewriting sentences for conciseness and clarity, for instance: “The findings from this study are organized into two main parts: (1) the performance of ChatGPT on RNLE, compared to the pass rate of nursing graduates; and (2) the evaluation of the limitations and advantages of using ChatGPT.” 3. Address inconsistencies in verb tenses, e.g., “ChatGPT may confuse or misunderstand…” instead of “ChatGPT may confusing or misunderstand…” 4.Clearly articulate the implications of ChatGPT’s limitations, offering insights into potential improvements and future research directions. 5.When mentioning other studies, ensure clear and concise integration of their findings, comparing and contrasting effectively with the current study’s results.
|
The discussion has been revised. |
|
4 |
Conclusion: The first sentence could be made more concise, e.g., “This study aimed to understand ChatGPT's performance on RNLE.” Avoid vague phrases like “may have potentially” – instead, be more definite: e.g., “ChatGPT has the potential…” Clearly articulate the practical implications and future applications of using ChatGPT in nursing education, ensuring that these are grounded in the study's findings. “Using ChatGPT… should be approached with caution, particularly due to hallucination misleading the answer, or regarding issues like plagiarism in article writing and homework assignments,” could be rephrased for clarity and conciseness, e.g., “Caution should be exercised when using ChatGPT, particularly due to potential inaccuracies in responses and concerns regarding plagiarism in written assignments.” |
The conclusion has been revised.
|
|
5 |
Proofread for grammar and consistency in verb tense and terminology. |
The English has been revised. |
Reviewer 3 Report
Comments and Suggestions for Authors
This is an important and interesting article. There is lack of or insufficient studies on this topic because it is a new concept especially in nursing education. The study is very original and innovative.
The method is very appropriate for this study.
My decision was based on issues related to sentence structure, grammar and punctuations.
I understand that author have challenges where English could be the secondary language and as such my recommendation to use an English editor to review and edit this article.

English language seems to be the major problem and there is need to employ the use of an editor who is fluent in English Language to revise and revamp this work. This will improve readability. Please delete the 2nd sentence on lines 7-8. It is a repetition.
Author Response
Thank you for the opportunity to re-submit our manuscript, titled “Performance of ChatGPT on Registered Nurse License Exam in Taiwan: A Descriptive Study”. We have carefully reviewed the reviewers’ comments and made revisions accordingly. Our response to reviewers’ comments is attached, with the revised portion highlighted in red. Please contact me if you have questions or need additional information. Thank you very much.
|
1 |
English language seems to be the major problem and there is need to employ the use of an editor who is fluent in English Language to revise and revamp this work. This will improve readability. Please delete the 2nd sentence on lines 7-8. It is a repetition. |
The information has been revised in the manuscript. |
Reviewer 4 Report
Comments and Suggestions for Authors
This study quantitatively and qualitatively evaluates the performance of the ChatGPT on the Taiwanese Registered Nurse License Exam.
This is a novel study in an area where knowledge is scarce.
On the other hand, the results were mediocre.
ChatGPT may be useful for faster responses and knowledge acquisition in Exam.
ChatGPT fails to address important aspects of nursing care, such as human creativity, critical thinking, and clinical reasoning.
These findings are, to some extent, to be expected.
However, they are new findings, even if mediocre. This study is worthwhile.
I believe that this paper would be much better if the following four points were improved.
1.In the Results section, the first line states.
"This study was to understand the performance of ChatGPT on the registered nurse license exam (RNLE) and evaluate the limitations and advantages using ChatGPT."
This should be described in the objectives section.
2.Table 1 lists the performance of the candidate and Table 2 lists the performance of ChatGPT. Table 1 lists the Pass percentage and Table 2 lists the Correct percentage.
Is it possible to compare, for example, a candidate's correct percentage with the correct percentage on the ChatGPT?
Even if it is difficult to compare them, it is not a problem because the title of this study states that it is a Descriptive Study and the authors are not conducting a comparative study.
3.In Table 1 and 2, I don't think it makes sense to list the p-values unless there is a full discussion of the questions in the Registered Nurse License Exam that the ChatGPT is good at or bad at.
4. If the word limit of the paper permits, more example questions and answers in Table 3-5 would add value to this paper.
Author Response
Thank you for the opportunity to re-submit our manuscript, titled “Performance of ChatGPT on Registered Nurse License Exam in Taiwan: A Descriptive Study”. We have carefully reviewed the reviewers’ comments and made revisions accordingly. Our response to reviewers’ comments is attached, with the revised portion highlighted in red. Please contact me if you have questions or need additional information. Thank you very much.
|
1 |
In the Results section, the first line states. "This study was to understand the performance of ChatGPT on the registered nurse license exam (RNLE) and evaluate the limitations and advantages using ChatGPT." This should be described in the objectives section. |
The result has been revised in the manuscript. |
|
2 |
Table 1 lists the performance of the candidate and Table 2 lists the performance of ChatGPT. Table 1 lists the Pass percentage and Table 2 lists the Correct percentage. Is it possible to compare, for example, a candidate's correct percentage with the correct percentage on the ChatGPT? |
It is difficult to compare them, because no data released from government. We try to revise more perfectly. |
|
3 |
Even if it is difficult to compare them, it is not a problem because the title of this study states that it is a Descriptive Study and the authors are not conducting a comparative study. |
It is difficult to compare them, because no data released from government. We try to revise more perfectly. |
|
4 |
In Table 1 and 2, I don't think it makes sense to list the p-values unless there is a full discussion of the questions in the Registered Nurse License Exam that the ChatGPT is good at or bad at. |
The tables have been revised. |
|
5 |
If the word limit of the paper permits, more example questions and answers in Table 3-5 would add value to this paper. |
The tables have added more information. |
Reviewer 5 Report
Comments and Suggestions for Authors
Dear author:
Abstract: Acronyms are generally not used in the abstract (the final decision to keep or remove them will be up to the editor); in any case, you should check not to duplicate the explanation of acronyms, as described in the first line of the abstract for AI. The sections should not be numbered in the abstract.
According STROBE statement, in the abstract methods section you should provide an informative and balanced summary of what was done
and what was found (Populations, sample, inclusion/exclusion criteria, variables, instruments, analysis process, and ethical considerations).
Keywords: ChatGPT is not a MesH term (because it is a novel topic, it is possible that Artificial Intelligence is correct to describe ChatGPT, but its use as a Keyword is an editorial decision). "Registered nurse license" is also not a MesH (in this case the MesH descriptor is "Nurse".
The proper Mesh for "nursing graduate" is "Education, Nursing, Graduate".
Introduction: Not to separate the introduction from the background at different sections. do not separate the introduction from the background at different points. Because it separates the sections, information is duplicated (e.g. when describing ChatGPT applications; and when describing the RNLE exam).
Do not duplicate the clarification of acronyms in the text (e.g. Registered Nurse License Exam (RNLE)).
Materials and methods: To clarify the methodology followed, it is necessary to organize its contents according to the STROBE statement (study design, setting, participants, variables, data sources/measurement, bias, stydy size, quantitative variables, statistical methods) available at the following link https://www.equator-network.org/reporting-guidelines/strobe/
The imput score and encoding methodology sections include information that is duplicated with the results (it is possible that part of this data corresponds to results). Although the subject matter is complex, you should limit yourself in the method to explaining what is going to be done and in the results to providing the rest of the information.
In tables, unify the use of punctuation marks for decimals. Only use the decimal point, do not use the comma. Table 3 and 4, corresponds better as a figure (improve the resolution of the text). Table titles should not include acronyms to be self-explanatory.
Conclusions: Should be limited to answering the objectives. In my opinion, part of the conclusions correspond to discussion.
Author Response
Thank you for the opportunity to re-submit our manuscript, titled “Performance of ChatGPT on Registered Nurse License Exam in Taiwan: A Descriptive Study”. We have carefully reviewed the reviewers’ comments and made revisions accordingly. Our response to reviewers’ comments is attached, with the revised portion highlighted in red. Please contact me if you have questions or need additional information. Thank you very much.
|
1 |
Abstract: Acronyms are generally not used in the abstract (the final decision to keep or remove them will be up to the editor); in any case, you should check not to duplicate the explanation of acronyms, as described in the first line of the abstract for AI. The sections should not be numbered in the abstract. |
The abstract have been revised. |
|
2 |
According STROBE statement, in the abstract methods section you should provide an informative and balanced summary of what was done and what was found (Populations, sample, inclusion/exclusion criteria, variables, instruments, analysis process, and ethical considerations). |
The study design, participants and setting all have been revised. |
|
3 |
Keywords: ChatGPT is not a MesH term (because it is a novel topic, it is possible that Artificial Intelligence is correct to describe ChatGPT, but its use as a Keyword is an editorial decision). "Registered nurse license" is also not a MesH (in this case the MesH descriptor is "Nurse". The proper Mesh for "nursing graduate" is "Education, Nursing, Graduate". |
The keywords have been revised. |
|
4 |
Introduction: Not to separate the introduction from the background at different sections. do not separate the introduction from the background at different points. Because it separates the sections, information is duplicated (e.g. when describing ChatGPT applications; and when describing the RNLE exam). Do not duplicate the clarification of acronyms in the text (e.g. Registered Nurse License Exam (RNLE)). |
The introduction have been revised. |
|
5 |
Materials and methods: To clarify the methodology followed, it is necessary to organize its contents according to the STROBE statement (study design, setting, participants, variables, data sources/measurement, bias, stydy size, quantitative variables, statistical methods) available at the following link https://www.equator-network.org/reporting-guidelines/strobe/ The imput score and encoding methodology sections include information that is duplicated with the results (it is possible that part of this data corresponds to results). Although the subject matter is complex, you should limit yourself in the method to explaining what is going to be done and in the results to providing the rest of the information. |
The study design, participants and setting all have been revised. |
|
6 |
In tables, unify the use of punctuation marks for decimals. Only use the decimal point, do not use the comma. Table 3 and 4, corresponds better as a figure (improve the resolution of the text). Table titles should not include acronyms to be self-explanatory. |
A table has been revised. |
|
7 |
Conclusions: Should be limited to answering the objectives. In my opinion, part of the conclusions correspond to discussion. |
The conclusions have been revised. |
Round 2
Reviewer 4 Report
Comments and Suggestions for Authors
The authors responded to the four or five comments from a reviewer to the best of his ability, and I believe the paper has improved.